EMBO
*reports*

# Molecular architecture of the N-type ATPase rotor ring from *Burkholderia pseudomallei*

Sarah Schulz[†], Martin Wilkes[†], Deryck J Mills, Werner Kühlbrandt [ID] & Thomas Meier[‡,*] [ID]

## Abstract

The genome of the highly infectious bacterium *Burkholderia pseudomallei* harbors an *atp* operon that encodes an N-type rotary ATPase, in addition to an operon for a regular F-type rotary ATPase. The molecular architecture of N-type ATPases is unknown and their biochemical properties and cellular functions are largely unexplored. We studied the *B. pseudomallei* $N_1N_o$-type ATPase and investigated the structure and ion specificity of its membrane-embedded c-ring rotor by single-particle electron cryo-microscopy. Of several amphiphilic compounds tested for solubilizing the complex, the choice of the low-density, low-CMC detergent LDAO was optimal in terms of map quality and resolution. The cryoEM map of the c-ring at 6.1 Å resolution reveals a heptadecameric oligomer with a molecular mass of ~141 kDa. Biochemical measurements indicate that the $c_{17}$ ring is $H^+$ specific, demonstrating that the ATPase is proton-coupled. The $c_{17}$ ring stoichiometry results in a very high ion-to-ATP ratio of 5.7. We propose that this N-ATPase is a highly efficient proton pump that helps these melioidosis-causing bacteria to survive in the hostile, acidic environment of phagosomes.

**Keywords** *Burkholderia pseudomallei*; c-ring stoichiometry; cryoEM; N-type rotary ATPase; proton homeostasis
**Subject Categories** Microbiology, Virology & Host Pathogen Interaction; Structural Biology

## Introduction

Rotary ATPases are dynamic membrane protein complexes, which play a crucial role in cellular bioenergetics, as they produce adenosine triphosphate (ATP) by exploiting the transmembrane electrochemical ion gradient. Working in reverse, they can pump ions against the gradient by using energy released by hydrolysis of ATP into adenosine diphosphate (ADP) and inorganic phosphate ($P_i$). Rotary ATPases share a common multiprotein subunit architecture forming two functionally tightly coupled parts, which act as two reversible motors [1,2]. The water-soluble subcomplex harbors three nucleotide conversion sites in the three catalytic β-subunits [3,4]. The membrane-embedded subcomplex consists of rotor and stator proteins, which use the proton- or sodium-motive force (*pmf, smf*) to translocate ions and drive ATP synthesis, or to pump ions in the opposite direction. Torque is transmitted from the membrane to the soluble part by the central stalk. Rotary ATPases can have up to three peripheral stalks, which are static and stabilize the complex during operation. F-type ATP synthases have a single outer stalk, while the A- and V-type ATPases have two or three [5]. Vacuolar (V-type) ATPases in eukaryotic cells work exclusively as proton pumps [6], while F- and A-type ATPases of bacteria or archaea can operate both in ATP synthesis and hydrolysis direction, depending on the physiological state of the cell. Recent advances in structural biology have provided remarkable new insights into previously unknown parts of F- and V-type rotary ATPases, especially the motor complex in the membrane [7–11].

The membrane rotors of rotary ATPases known to date consist of 8–15 copies of c-subunits in the c-ring [12,13]. The ring stoichiometry is determined by the amino acid sequence of the c-subunit [14–16]. The usual c-subunit found in most F-type ATP synthases is a simple α-helix hairpin. The c-subunit sequence does not only determine the stoichiometry but also the type of ion ($H^+$ or $Na^+$), translocated by the ATPase/synthase for energy conversion. It is the sole determinant in all ATP-dependent bioenergetics processes that decides whether ATP production in a given organism is powered by the *pmf* or the *smf*. The C-terminal α-helix of each c-subunit contains a conserved and functionally essential carboxylate (Glu or Asp), which lies at the outer surface of the c-ring in the membrane center. Ions bind to or are released from these sites through two conserved half channels from the cytoplasmic (matrix) side and from the cell exterior or lumenal side [17]. The c-ring stoichiometry defines the number of ions that are transported along this pathway and hence the ion-to-ATP ratio in rotary ATPases/synthases of any given species [13,14,18].

An *in silico* study of the distribution of rotary ATPase operons identified a novel *atp* operon in several phylogenetically independent groups of bacteria and archaea [19]. The operon was always found in addition to other F-/A-type *atp* operons in the same genome. The protein complexes encoded by these so-called "*novel*"

Department of Structural Biology, Max Planck Institute of Biophysics, Frankfurt am Main, Germany
*Corresponding author. Tel: +44 2075943056; E-mail: t.meier@imperial.ac.uk
[†]These authors contributed equally to this work
[‡]Present address: Department of Life Sciences, Imperial College London, London, UK

or "next to" *atp* operons are referred to as N-type ATPases and were predicted to be "predominantly $Na^+$-selective" [19]. A later study of the halotolerant alkaliphilic cyanobacterium *Aphanothece halophytica* showed that, besides the conventional (F-type) $H^+$-ATP synthase, this bacterium indeed expresses a $Na^+$-coupled rotary ATPase of the N-type [20], suggesting a role in counteracting salt stress. Apart from this report, no structural or biochemical data are available of any N-type ATPase. In particular, the stoichiometry and structure of their bioenergetically decisive c-rings are unknown.

We undertook to investigate the structure of the N-type ATPase rotor ring from the Gram-negative, pathogenic β-proteobacterium *Burkholderia pseudomallei*, the causative agent of pneumonia-causing melioidosis/glanders [21–23]. Using single-particle cryoEM, we obtained a map at 6.1 Å resolution that reveals the basic architecture of an N-type rotor ring composed of 17 identical c-subunits. Biochemical evidence indicates that the c-subunits of this rotor predominantly bind $H^+$, indicating that physiologically, the *B. pseudomallei* N-type ATPase is proton-coupled.

## Results and Discussion

### Expression of the *Burkholderia pseudomallei* N-type ATPase and purification of the c-ring

The N-type *atp* operon of *B. pseudomallei* has nine open reading frames (Fig 1A). The order of appearance of the genes is similar to conventional bacterial F-type *atp* operons [24], except that *atpD* (β-subunit) and *atpC* (ε-subunit) are shifted to the front of the operon. The translated amino acid sequences indicate a close relationship to F-ATP synthase subunits, with the exception of *atpQ*, which encodes putative a membrane protein of yet unknown function. The gene encoding the 8.3 kDa c-subunit (*atpE*) is located in the central region of the N-ATPase operon. Based on the sequence alignment with known F-type ATP synthase c-subunits, it is predicted to form a hairpin of two transmembrane helices that contains two carboxylates at the conserved ion-binding site (Fig 1B), similar to other $Na^+$- and $H^+$-binding c-rings [25,26].

To show that the genes of the N-type ATPase encode a real protein complex and to produce the complete N-type ATPase of *B. pseudomallei*, we cloned the entire N-*atp* operon into a pHERD28T *Burkholderia* expression vector system [27]. The vector pHERD_BPN1No carrying the complete set of N-ATPase subunits was expressed in the non-pathogenic host *Burkholderia thailandensis*. Western blot analysis of *B. thailandensis* membranes with a c-subunit-specific antibody gave a strong signal for a c-ring oligomer of the same molecular mass as the c-ring complex that was isolated from *Escherichia coli* cells expressing only the *B. pseudomallei* c-subunit. This indicates that the *B. pseudomallei* c-oligomer does not alter its stoichiometry when it is expressed either by itself or together with all other N-type ATPase subunits in an expression host (Fig 2A and B). This observation is in line with our earlier work on the heterologous expression of c-rings from *Ilyobacter tartaricus* or *Caldalkalibacillus thermarum* TA2, indicating that the c-ring stoichiometry is conserved in host cells and does not depend on external factors such as lipid composition, expression medium, or growth temperature [28–30].

Negative-stain electron microscopy (EM) was performed with inverted membrane vesicles of *B. thailandensis* cells expressing the *B. pseudomallei* N-type ATPase, taking advantage of an anti-His$_6$-tag on the β-subunit (*atpD*) of the soluble $N_1$ headgroup (the equivalent of the $F_1$-$\alpha_3\beta_3$ head in F-type ATP synthase) to attach an immunogold label for visualization. Gold beads were found on the outside of the vesicles only after N-type ATPase induction (Fig 2C), showing that the *B. pseudomallei* N-type rotary ATPase β-subunit was successfully expressed and the $N_1N_o$ ATPase was assembled and incorporated into the membrane of *B. thailandensis* host cells.

Following our previously established strategy for expressing c-subunits and their assembly into c-rings, we used a heterologous *E. coli* expression system [28,30]. *Burkholderia pseudomallei atpE* was cloned into a T7 expression vector and the c-subunit was expressed in the *E. coli* strain Lemo21. The c-subunit did not have an affinity tag, and we used a purification method that bases on ammonium sulfate precipitation [31]. SDS-gel electrophoresis indicated a band at ~90–100 kDa. The SDS-stable oligomeric c-ring complex dissociated into monomeric subunits upon acid treatment (Fig 2A). The subunit mass was determined by MALDI-MS as 8,349.6 kDa for the unmodified protein and 8,378.3 kDa for the formylated protein, in good agreement with predictions (Fig EV1). The monomer mass together with the slow migration of the oligomer on the SDS–polyacrylamide gel suggested an unusually high c-ring stoichiometry for the *B. pseudomallei* N-type c-ring.

### Ion specificity

To address the ion specificity of the isolated *B. pseudomallei* c-ring oligomer, the detergent-solubilized complex was exposed to *N*-cyclohexyl-*N'*-(4-(dimethylamino)-α-naphtyl)-carbodiimide (NCD-4), a fluorescent analogue of the ATP synthase inhibitor *N,N'*-dicyclohexylcarbodiimide (DCCD) (Figs 3 and EV2). NCD-4 reacts covalently with the protonated, ion-binding carboxylates of c-rings. Therefore, it can be used to directly characterize the type of ion that is bound in a given ATP synthase by measuring a continuous increase of fluorescence upon addition of the fluorophore. For example, in the case of the $Na^+$ binding c-ring from *I. tartaricus*, the reaction can be specifically and immediately stopped by adding 15 mM NaCl resulting in 1% remaining labeling activity [31]. Similarly, in a $H^+$ binding c-ring, the increase of fluorescence is completely stopped (no measurable remaining activity) by deprotonation of the ion-binding carboxylates, caused by the shift of pH to 9.0 in the cuvette [32]. For testing our *B. pseudomallei* c-oligomer, we chose to initiate the reaction at pH 6.0 at which the ion-binding c-oligomer carboxylates are still protonated and is able to react rapidly with NCD-4, while all water-exposed carboxylates (e.g., loop region, Fig 1B) are deprotonated and are not involved in the reaction. A continuous increase of fluorescence was monitored for several minutes and then quantified. The reaction starts with a linear increase of fluorescence (Fig 3A, blue arrow). We then tested the effect of $Na^+$ (Fig 3A, red arrow) and depletion of $H^+$ (pH 9.0) on the reaction efficiency (Fig 3B, red arrow). $Li^+$ and $Cs^+$ ions were used as controls to test the effect of other, non-physiological monovalent cations on the labeling reaction (Fig EV2). The addition of 15 mM $Na^+$ reduced NCD-4 labeling from the initial rate (defined as 100%) to 40% remaining labeling activity. The effect of 15 mM

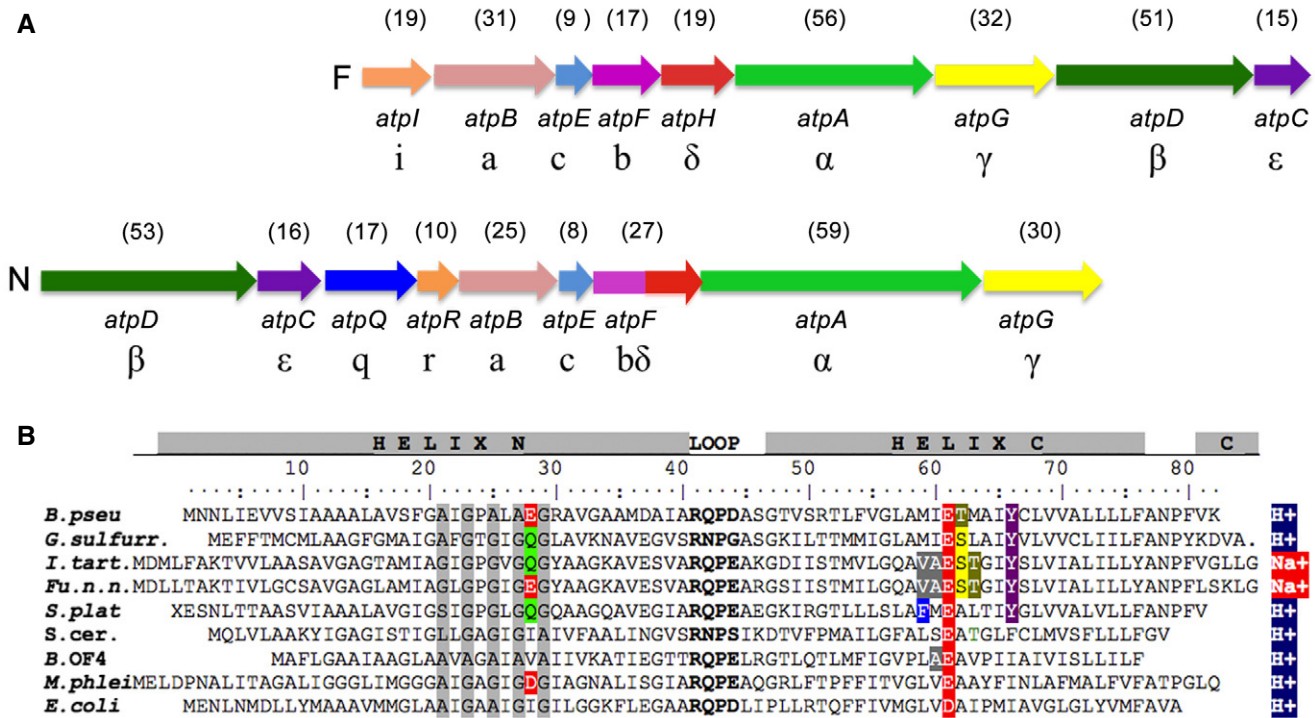

**Figure 1. N-*atp* operon and c-subunit sequence of *Burkholderia pseudomallei* aligned with other known c-subunits.**

A   Gene order of the F-type *atp* operon (top) compared to the N-type *atp* operon (bottom). In the N-*atp* operon, the *atpD* and *atpC* precede *atpB* and are followed by *atpQ* and *atpR*, which are absent in the F-*atp* operon. Arrows indicate coding sequences. The theoretical molecular masses of the F-ATP and the N-ATPase gene products are given in kDa (brackets) above each gene. In *B. pseudomallei*, the F-type operon is located on the genomic chromosome 1 and the N-type operon is located on genomic chromosome 2, flanked by virulence-specific genes [22].

B   Amino acid alignment of selected H+- and Na+-specific F-type c-subunits with the N-type c-subunit of *B. pseudomallei*. The conserved loop region is bold and conserved residues involved in ion binding are colored. The conserved glycine motif as discussed in the text is highlighted in gray. Species are as follows: *Burkholderia pseudomallei* (numbering), *Geobacter sulfurreducens, Ilyobacter tartaricus, Fusobacterium nucleatum* subsp. *nucleatum, Spirulina platensis, Saccharomyces cerevisiae, Bacillus pseudofirmus* OF4, *Mycobacterium phlei,* and *Escherichia coli*.

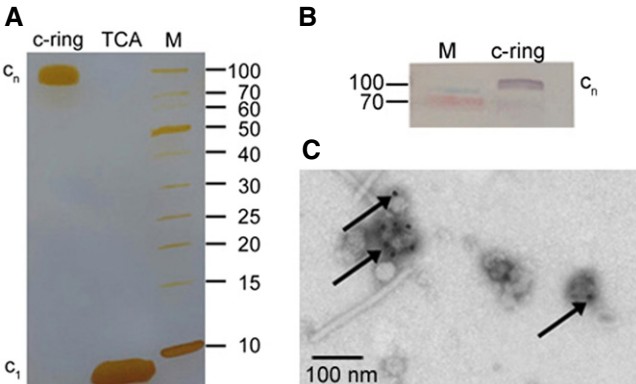

**Figure 2. Oligomeric size of the *Burkholderia pseudomallei* N-type ATPase rotor ring.**

A   SDS–PAGE of the isolated *B. pseudomallei* c-ring before and after TCA treatment, which dissociates the complex into monomers.

B   Western blot of the *B. pseudomallei* N-ATPase in *Burkholderia* membranes with a specific anti-c-subunit antibody. The rotor ring shows the same oligomeric size as the isolated complex in (A).

C   Negative-stain electron microscopy of the *B. pseudomallei* N-ATPase β-subunit with immunogold-labeled His$_6$-tag in inverted membrane vesicles.

Li$^+$ (55% remaining) and Cs$^+$ (44% remaining) was similar, while a pH increase to 9.0 (= complete deprotonation of c-ring glutamates) again resulted in an immediate and complete stop of the reaction with only 1% of rest labeling. A further increase of the Na$^+$, Li$^+$, and Cs$^+$ concentration (150 mM) to a value similar to a physiological concentration of blood serum decreased the total NCD-4 labeling efficiency to 27, 17, and 24%, respectively (Figs 3A and EV2). A quantification of the labeling efficiencies is given in Table EV1. Note that the rather high increase of cationic strength in the measurement causes effects on fluorescence quenching of the fluorescent probe, as it has been described previously [33]. In addition, the combined effects of salt on the availability of water and the critical micelle concentration of the detergent need to be taken into consideration. Importantly however, the observation contrasts with the strong and immediate effect of Na$^+$ on NCD-4 labeling of Na$^+$ binding c-rings from *I. tartaricus* [31], while it matches that observed with a c-ring isolated from a H$^+$ ATP synthase remarkably well [32]. In contrast to what had been predicted from the polypeptide sequence [19], these biochemical data suggest that the *B. pseudomallei* N-type ATPase c-ring is predominantly H$^+$ selective, indicating it is coupled to protons and hence contributes to or utilizes the proton gradient, not the sodium gradient, across the bacterial membrane.

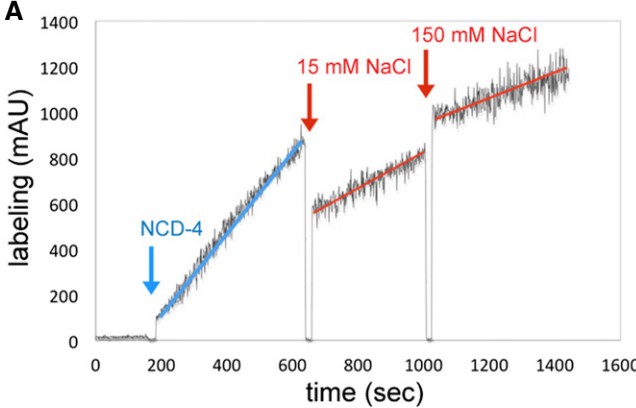

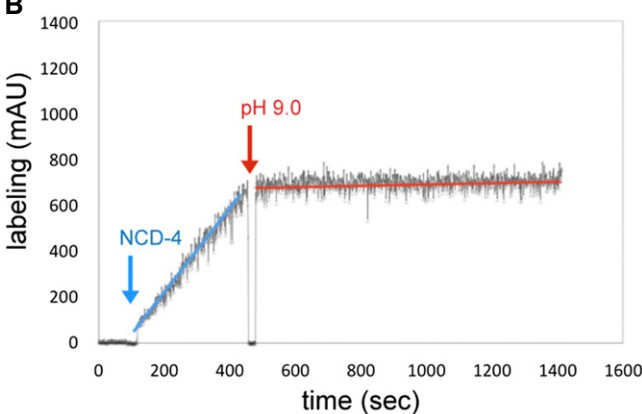

**Figure 3.  Effect of Na⁺ and high pH on the kinetics of the NCD-4 modification of the *Burkholderia pseudomallei* N-type ATPase rotor ring.**

100 μM NCD-4 was added (blue arrows) to the purified c-ring in 1.0% DDM and 2-(*N*-morpholino)ethanesulfonic acid (MES) buffer, pH 6.0. Continuous increase of fluorescence upon NCD-4 binding to Glu61 of the c-ring.

A  Addition of 15 and 150 mM NaCl (red arrows) decelerated the reaction due to unspecific salt effects.

B  Increasing the pH to 9.0 by adding 2.5 M Tris base stock solution (red arrow) abolished NCD-4 binding to Glu61. Results of typical experiments are shown. The experiments were repeated three times each.

## Single-particle cryoEM

We proceeded to determine the structure of the rotor ring by single-particle cryoEM. Like some other rotor rings [14], the *B. pseudomallei* c-ring complex proved to be stable in many solubilizing agents tested. We investigated the c-ring in the three detergents lauryl dimethyl amineoxide (LDAO), dodecyl maltoside (DDM), octaethylene glycol monododecyl ether ($C_{12}E_8$), and the amphipathic polymer surfactant amphipol A8-35 [34]. The choice of surfactant proved to be a decisive factor for the success of the analysis and the map resolution obtained. There was a clear correlation between the density of the detergent or amphipol, micelle size, and the resolution achieved (Table EV2).

Notwithstanding the protein complex mass of 141 kDa, which is low for cryoEM, the c-ring complex was visible in thin layers of amorphous ice (Fig EV3A). A total of 213,000 particles in LDAO, 76,000 in DDM, 115,000 in $C_{12}E_8$, and 226,000 particles in amphipol

A8-35 were picked semi-automatically. In 2D class averages, top views of rings in LDAO or DDM showed 17 density characteristics of transmembrane α-helices in projection (Fig 4A and B). In LDAO, two concentric rings of 17 staggered α-helix densities were resolved clearly (Fig 4B). In $C_{12}E_8$, the 17 α-helix densities were poorly resolved (Fig 4C), and in amphipol A8-35, they were not resolved at all (Fig 4D), even though the largest number of particles contributed to the average. Transmembrane helices also stood out in side views of c-ring complexes solubilized in DDM and especially LDAO, whereas this was not the case in $C_{12}E_8$ or amphipol A8-35.

The 2D class averages of the top and side views explain why LDAO works best. Due to the small diameter of its micelle, LDAO covers only about half the length of the transmembrane ring helices. The large difference in density between protein and amorphous ice (Table EV2) ensures high contrast of the solvent-exposed part of the protein, which enables more accurate particle alignment during image processing and hence, higher resolution. Moreover, the density of LDAO is closer to that of amorphous ice than the other surfactants (Table EV2). As a result, contrast matching between solvent and protein [35] in LDAO is minimal and protein contrast is highest, which again helps the alignment accuracy.

3D refinement and particle polishing [36] with the best 2D class averages of LDAO-solubilized complexes resulted in a 6.1 Å map of the rotor ring (Figs EV3B and 4E and F). The individual α-helices, which form the 17 identical c-subunits were well-resolved. The c-ring of the *B. pseudomallei* N-type ATPase is a cylinder with a height of ~65 Å. Its outer diameter is ~70 Å at the periplasmic side, ~58 Å in the center of the lipid bilayer, and ~65 Å at the cytoplasmic side, giving the rotor a slight hourglass shape that is characteristic of many c-ring rotors of F- and V-type ATPases [37,38] (Movie EV1). Its hydrophobic central cavity with a diameter of ~30 Å is presumably filled with lipid [39].

## Model building and structure analysis

Seventeen copies of a helix hairpin with the *B. pseudomallei* polypeptide sequence modeled on the *Fusobacterium nucleatum* c-ring structure [26] were fitted to the EM map and refined against the cryoEM density using real-space-refinement in Phenix (Fig 4F). The fit of the $C_\alpha$ chain was consistent with the conserved RQPD sequence in the loop connecting the two helices and the conserved pattern of glycine/alanine residues in the N-terminal helix at the interface to the next c-subunit in the ring (Fig 1B). It has been shown that if some of the conserved glycine residues are changed to alanine, the helical packing changes, which can result in higher c-ring stoichiometries [16,32,40,41]. The A×G×A×A×G sequence in the N-terminal helix of the *B. pseudomallei* c-ring thus contributes to helix packing and to the unusually high $c_{17}$ stoichiometry.

### Ring architecture

Overall dimensions of the *B. pseudomallei* c-ring closely resemble those of the 15-subunit *Spirulina platensis* c-ring [42], even though the former has two more subunits. The inner N-terminal α-helices are mostly straight, bending only very slightly toward the central hydrophobic cavity. The outer, C-terminal helices are S-shaped, with a slight kink near the ion-binding site (Glu61) and a sharper kink at Phe76 on the periplasmic membrane surface. On the cytoplasmic side, Ser50 marks the boundary of the hydrophobic,

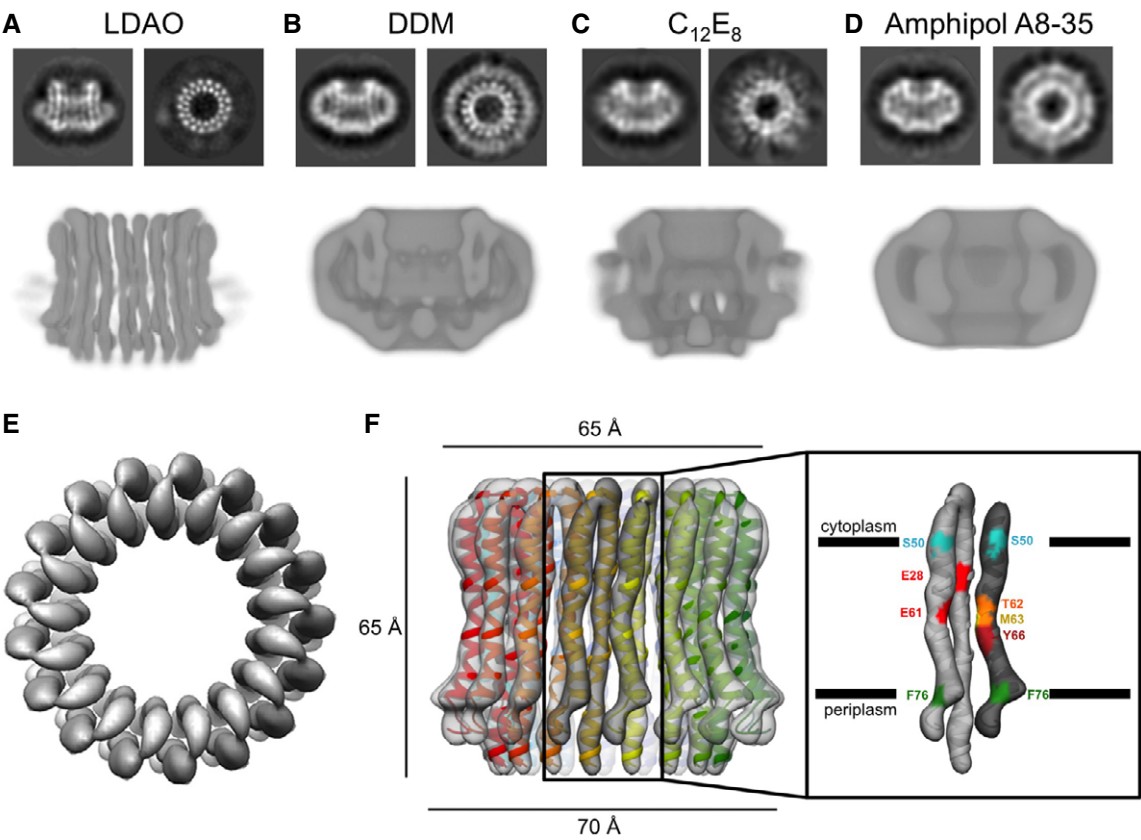

**Figure 4.  Electron cryo-microscopy of the *Burkholderia pseudomallei* N-type ATPase rotor ring.**

A–D   Representative 2D class averages of the rotor ring showing the ring from the side and top as well as slices through 3D maps after refinement for (A) LDAO, (B) DDM, (C) $C_{12}E_8$, and (D) amphipol A8-35.

E   Top view of the rotor ring in LDAO with 17 identical hairpin-like c-subunits fitted.

F   Side view of the c-ring in LDAO with the refined model fitted. The helices in the outer ring are slightly shorter than the inner helices. Two adjacent c-subunits form a functional ion-binding unit. The positions of residues involved in ion binding or defining the membrane boundary are shown in different colors.

membrane-embedded region, indicating a lipid bilayer thickness of 35 Å (Fig 4F).

### Ion-binding site

The ion-binding Glu61 is located near the center of the lipid bilayer. In our *B. pseudomallei* model, its carboxylate sidechain points outward, as in the X-ray structures of several F-type ATPase rotor rings [14]. This acidic residue, which is crucial for activity in all rotary ATPases, is surrounded by sidechains involved in $Na^+$ or $H^+$ coordination [32,37,42]. A second glutamate (Glu30) in the N-terminal helix points toward the ion-binding site, reminiscent of c-ring structures with a two-carboxylate ion-binding motif [25,26]. This site has no apparent influence on the NCD-4 modification reaction (Fig 3). Therefore, as in other c-rings with the two-carboxylate motif, this second glutamate is protonated at all times and serves as a hydrogen-bond donor. Finally, Thr62, Met63, and Tyr66 in the C-terminal helix are likely to contribute to the hydrogen-bonding network around the proton-binding site. Even though, at 6.1 Å, the map does not resolve sidechains, it is safe to state that the c-rings of N-type and F-type ATPases are very similar with respect to overall architecture, including the layout of their ion-binding sites, allowing us to conclude that both types work essentially in the same way.

### Analysis of $N_o$ subunits in contact with the c-ring based on sequence comparison

We compared the polypeptide sequences of other subunits in the $N_o$ subcomplex that are homologous to subunits γ, ε, and a (Fig 1A), which, in F-type ATPases, are close to or in direct contact with the c-ring. As expected, critical residues, in particular the functionally essential Arg164 in subunit a [43] and the c-ring contacting His40 in ε [44] are conserved, as is the overall pattern of hydrophobic and polar sidechains (Fig EV4). Notable exceptions are the residues at the interface between subunit γ and the c-ring, in particular the conserved motif at Tyr207/Glu208 (Fig EV5). In *B. pseudomallei*, this residue pair is exchanged against Val/His, and this is likely to affect the c-ring/γ interaction. In general though, the larger diameter of the c-ring and higher c-subunit stoichiometry of *B. pseudomallei* are not reflected in the primary subunit sequences. We conclude that not only the amino acid sequences but also the structures of the neighboring subunits in the $N_o$ subcomplex are overall conserved and similar to those of the conventional F-type ATPases. This includes in particular the lengths of helices 5 and 6 of the a-subunit that wrap around the c-ring [7–11], which can thus adapt to the larger diameter of the *B. pseudomallei* $c_{17}$ ring.

## Conclusions

We determined the cryoEM structure of a c-ring of an N-type ATPase from the bacterium *B. pseudomallei*. The c-ring has an unexpected heptadecameric $c_{17}$ subunit stoichiometry. This finding expands the range of known c-ring stoichiometries that so far extended from 8 [13] to 15 [12], and refutes an earlier speculation that N-type c-rings would turn out to be on the small side of the range [19].

Each c-subunit in the *B. pseudomallei* $c_{17}$ ring is a hairpin of two α-helices with an ion-binding site at the interface between adjacent c-subunits, as in the F-type ATPases. The c- and K-rings of the V-type ATPases from *Saccharomyces cerevisiae* and *Enterococcus hirae*, respectively, are even larger, equivalent to 20 single-hairpin c-subunits [10,38]. The K-ring from *E. hirae* only contains 10 ion-binding sites, owing to the fact that each K-subunit is a double-helix hairpin with only one single acidic ion-binding glutamate [38]. The

high degree of sequence similarity between N-type and F-type operons supports the notion that an $N_1N_o$-type ATPase shares the same structural features and functionalities of an $F_1F_o$-ATPase. However, given that the *B. pseudomallei* N-type ATPase can translocate 17 ions per complete rotation, this results in an unusually high ion-to-ATP ratio of 5.7, making it a highly efficient proton pump.

It has been reported that during the early stages of infection that causes melioidosis, *B. pseudomallei* can exist in acidic, membrane-bound compartments such as polyphagosomes [45,46]. As the *B. pseudomallei* genome also encodes a conventional F-type ATP synthase, we propose that the N-type rotary ATPase, as a highly efficient ATP-driven proton pump, enables the bacterium to cope with acid stress and maintain pH homeostasis to survive in this hostile environment, while the F-type ATP synthase generates ATP by oxidative phosphorylation (Fig 5). In support of this notion is also the fact that the N-type operon is flanked by genes on chromosome 2 that are involved in the bacterium's virulence, while the

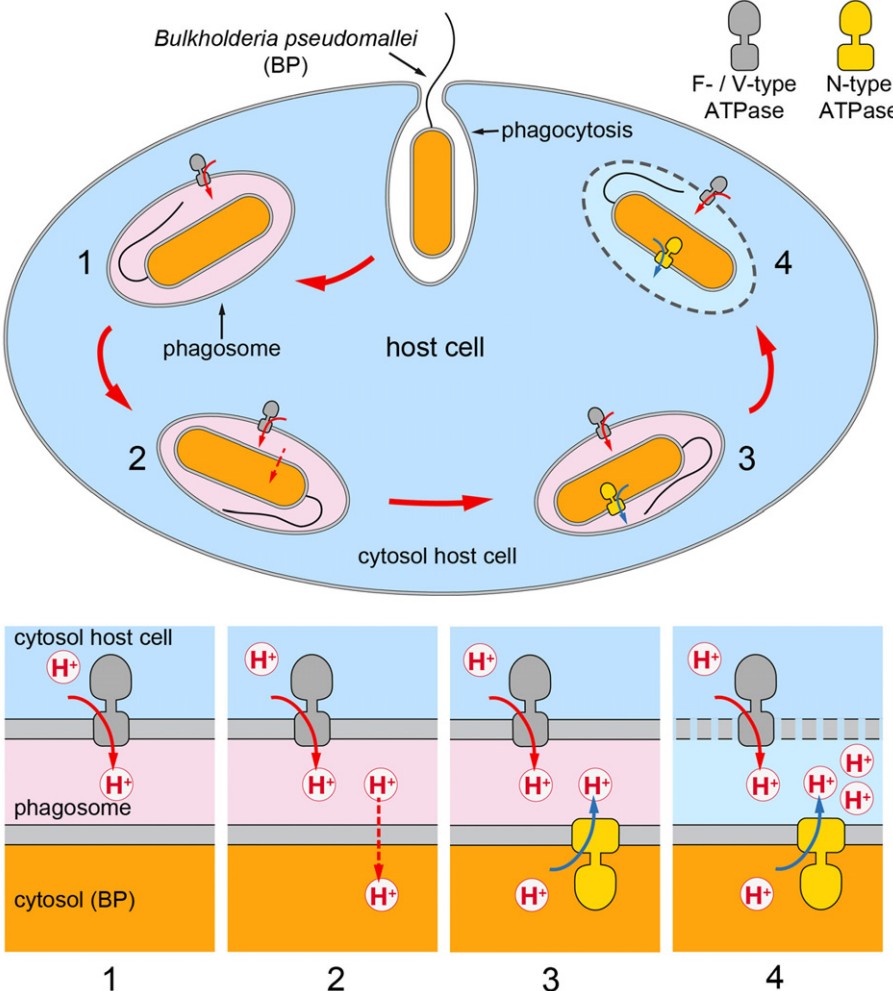

**Figure 5.  Proposed role of the *Burkholderia pseudomallei* N-type ATPase in H⁺ homeostasis and phagosome escape during the early state of infection.**
*Burkholderia pseudomallei* enters macrophage phagosomes that are acidified by a host cell-specific V-type ATPases (1). Due to the high pH gradient, protons leak through the cell membrane of *B. pseudomallei* and acidify the *B. pseudomallei* cytosol (2). The proposed role of the N-type ATPase is to redress the internal pH by pumping protons out of the bacterial cell and to maintain protein homeostasis (3). *B. pseudomallei* expresses secretion apparatus (bsa) system, which is essential for phagosome escape (4). Drawing adapted from Wiersinga *et al* [23].

F-type operon resides on chromosome 1 of the *B. pseudomallei* genome (Fig 1) [22]. Furthermore, we also speculate that the $c_{17}$ ring in the N-ATPase may also play a role in $H^+$ homeostasis of the bacterial cell during infection, enabling the cell to express a secretion apparatus, which is essential for its phagosome escape [23] (Fig 5, stage 4). Future work is required to address the role of N-type ATPases in cell homeostasis and pathogenicity in more detail.

# Materials and Methods

### Cloning of pHERD28T_atpBP

The N-*atp* operon was amplified from genomic DNA of *B. pseudomallei* by four different PCRs. Due to the high GC content (70%), Extender Polymerase (5 Prime) and a combinatorial enhancer solution (CES) were used [47]. Four fragments were created containing the restriction sites *Bgl*II and *Nde*I (fragment 1: 2,035 bp), *Nde*I and *Apa*I (fragment 2: 1,755 bp), *Apa*I and *Ssp*I (fragment 3: 1,548 bp), and *Ssp*I and *Hind*III (fragment 4: 2,106 bp). The fragments were blunt-end-cloned into a pJET1.2 kit for amplification (Fermentas), restricted, and fragments 1 and 2 were assembled with fragments 3 and 4 by simultaneous cloning into a pET24a vector, which was digested with *Bgl*II and *Apa*I or *Apa*I and *Hind*III, respectively. By restriction of pET24a_1+2 and pET14a_3+4, two new larger fragments (3,738 bp and 3,596 bp, respectively) were obtained and simultaneously cloned into a pBAD_HisB vector digested with *Bgl*II and *Hind*III. Subsequently, the entire N-*atp* operon (7,441 bp) was digested from the pBAD_HisB vector by *Nco*I and *Hind*III and ligated in a digested pHERD28T vector resulting in the pHERD_BPN1No vector for expression in *B. thailandensis* cells under control of the pBAD promotor.

### Expression and purification of the *Burkholderia pseudomallei* N-type ATPase c-ring

The *B. pseudomallei* N-type ATPase c-ring was heterologously expressed in *E. coli* Lemo21 using a pt7-7 expression vector [48]. Protein expression in 2 l LB medium was induced with 0.5 mM IPTG at $OD_{600}$ of 0.8 and cells were harvested after 4 h of incubation at 37°C under continuous shaking at 120 rpm. Cells were harvested by centrifugation at 5,000 *g* and 4°C for 30 min and resuspended in 10 ml/l cell culture of 50 mM potassium phosphate buffer (KPB) pH 8.0. Cells were disrupted by several passages through a micro-fluidizer at 4°C and 1,200 bar in the presence of 1 mM Pefablock (Sigma-Aldrich, D), 1 mM dithiothreitol (DTT), and traces of deoxyribonuclease A (DNase A). Cell debris was removed by centrifugation for 35 min at 24,000 *g* and 4°C. Membranes were collected at 230,000 *g* for 60 min at 4°C and resuspended in 50 mM KPB pH 8.0, 100 mM NaCl, and 10% (v/v) glycerol. The c-ring was extracted from *E. coli* membranes with 2% (w/v) *N*-lauroylsarcosine for 12 min at 66°C. Contaminating proteins solubilized with the c-ring were salted out with 70% saturated $(NH_4)_2SO_4$ in 50 mM Tris–HCl pH 8.0 buffer. The c-ring was dialyzed against 20 mM Tris/HCl with a pH adjusted to 8.0 at 22°C until the protein precipitated. The c-ring precipitate was collected by centrifugation and resuspended in 3% (w/v) *n*-dodecyl-β-D-maltoside (DDM) and 20 mM

Tris/HCl pH 8.0. The protein was further purified by size-exclusion chromatography (Äkta Purifier) in 20 mM Tris/HCl pH 8.0, 150 mM NaCl, 0.05% (w/v) DDM on a Superose 6 column (GE Healthcare, Munich, D). The eluted protein was concentrated by centrifugation, and the final protein concentration was determined by the BCA assay (Pierce).

### NCD-4 modification reaction of the *Burkholderia pseudomallei* c-ring

NCD-4 modification was performed with a 60 µl sample containing 0.45 mg/ml c-ring in 20 mM Tris/HCl pH 8.0 and 1.5% (w/v) DDM adjusted to pH 6.0 by the addition of 0.5 M 2-(*N*-morpholino)ethane sulfonic acid (MES)/HCl, pH 5.0. The reaction was started with 100 µM NCD-4 (Invitrogen) prepared from a stock solution in 10% (w/v) DDM. The continuous increase of fluorescence was monitored in an F-450 Hitachi Fluorescence Spectrophotometer ($\lambda_{ex}$ = 324 nm, $\lambda_{em}$ = 440 nm) at 25°C.

### Expression, Western blot analysis, and immunogold labeling of the *Burkholderia pseudomallei* N-type ATPase

Vector pHERD_BPN1No was transformed in *B. thailandensis* E264 for quasi-homologous expression of the N-type ATPase. Protein expression was induced with 0.2% arabinose at $OD_{600}$ of 0.6 and cells were harvested after shaking at 120 rpm for 4 h at 37°C. Cells were harvested at 5,000 *g* and 4°C for 30 min and resuspended in 10 ml/l culture of 20 mM Tris/HCl buffer pH 8.0. Membrane vesicles were prepared by several passages through a French press at 4°C and 1,000 bar in the presence of 1 mM Pefablock, 1 mM DTT and traces of deoxyribonuclease A. Cell debris was removed by centrifugation at 24,000 *g* and 4°C for 35 min. Membrane vesicles were harvested at 230,000 *g* and 4°C for 60 min and resuspended in 20 mM Tris/HCl buffer pH 8.0, 100 mM NaCl, 5% (v/v) glycerol, 5 mM $MgCl_2$, and 0.1 mM Pefablock.

Western blot analysis of the expression was performed with a specific antibody against the ATPase c-subunit (ATP synthase subunit c, AS09591 Agrisera Antibodies) and an alkaline phosphatase-conjugated second antibody (anti-mouse IgG, A4312 Sigma-Aldrich). Membrane vesicles were labeled with a specific antibody (monoclonal anti-polyhistidine antibody produced in mouse, H1029 Sigma-Aldrich) against the His-tag on the β-subunit of the N-type ATPase and a secondary antibody carrying a 8-nm gold particle. The labeled vesicles were deposited on a carbon-coated grid, negatively stained with 1% (w/v) uranyl acetate and examined in a CM120 electron microscope (Phillips).

### Electron cryo-microscopy

For electron cryo-microscopy, 3 µl aliquots of purified c-ring in amphipol A8-35, DDM, $C_{12}E_8$, or LDAO at a concentration of 2.5 mg/ml were loaded on glow-discharged holey carbon grids (Quantifoil R2/2). Grids were plunge-frozen in a Vitrobot (FEI) at 75% humidity and 10°C with blotting times ranging from 7 to 11 s. Micrographs were collected at liquid nitrogen temperature on a JEOL 3200 FSC electron microscope, operating at 300 kV with an in-column energy filter and equipped with a K2 Summit direct electron detector camera (Gatan). The slit width was adjusted to 20 eV. Images were

recorded in counting mode at a nominal magnification of 30,000×, corresponding to a pixel size of 1.14 Å at the specimen. The total exposure time was 5 s (amphipol A8-35, DDM and $C_{12}E_8$) or 7 s (LDAO) leading to an accumulated dose of 40–70 electrons per $Å^2$. Each image was fractionated into 25–35 subframes with an accumulation time of 0.2 s. Defocus values in the final datasets range from 1 to 4 μm.

**Image processing**

Global beam-induced motion of images was corrected with Motion-Corr [49], and the contrast transfer function for each image was determined with CTFFIND3 [50]. RELION-1.3 [51] was used for automated particle selection and manual deletion of false-positive and addition of false-negative particles. c-ring symmetry was determined by 2D classifications using Relion-1.3, XMIPP [52] and EMAN [53] with phase-flipped CTF correction. For 3D reconstruction, a starting model was generated in EMAN [53]. Unsymmetrized 2D and 3D classifications were used in Relion-1.3 to discard false-positive particles. A subsequent 3D classification was performed with one class, applying 17-fold symmetry at an angular sampling of 3.75°. The output of the 3D classification of particles in LDAO was used as a starting model for the final 3D autorefinement. Particle-based beam-induced movement correction and particle polishing were performed in Relion-1.3 [36,54] using a running average of seven movie frames and a standard deviation of one pixel for the translational alignments.

**Model building and refinement**

The WHATIF homology modeling server [55] was used to generate a model of the *B. pseudomallei* N-type ATPase rotor ring based on the *F. nucleatum* subsp *nucleatum* c-subunit [26]. The two c-subunit sequences are 46% identical and 76% of all residues are conserved; 17 copies of the homology model were fitted to the EM map in COOT [56] and Chimera [57]. The resulting structure was refined in PHENIX [58]. The EM data were deposited in the Electron Microscopy Data Bank (EMDB) with accession code EMD-3546.

**Expanded View** for this article is available online.

## Acknowledgements

We thank Herbert Schweizer and Linnell Randall from Colorado State University for providing the pHERD28T expression vector, Julian Langer (Max Planck Institute of Biophysics, Frankfurt) for MALDI-MS of the *B. pseudomallei* c-ring, and Friederike Joos for immunogold labeling of the N-type ATPase. This work was funded by the Max Planck Society, the Collaborative Research Centre (SFB) 807 of the German Research Foundation (DFG), the Cluster of Excellence Frankfurt "Macromolecular Complexes" (DFG Project EXC 115), and by the Wellcome Trust (WT110068/Z/15/Z).

## Author contributions

SS performed cloning, expression, biochemical experiments, and negative-stain EM. MW performed electron cryo-microscopy and data processing. DJM supported the cryoEM work and devised the data collection strategy. TM initiated and supervised the project. SS, MW, WK, and TM interpreted the data and wrote the paper.

## Conflict of interest

The authors declare that they have no conflict of interest.

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
