## [Review Process File · EMBO Reports]

Manuscript EMBO-2016-43374

Molecular architecture of the N type ATPase rotor ring from *Burkholderia pseudomallei*

Sarah Schulz, Martin Wilkes, Deryck J. Mills, Werner Kühlbrandt, and Thomas Meier

Corresponding author: Thomas Meier, Imperial College, London

Review timeline:

Submission date:	19 September 2016
Editorial Decision:	24 November 2016
Revision received:	20 December 2016
Editorial Decision:	11 January 2017
Revision received:	02 February 2017
Accepted:	09 February 2017

Editor: Achim Breiling

Transaction Report:

1st Editorial Decision

24 November 2016

Thank you for the submission of your research manuscript to EMBO reports. We have now received reports from the three referees that were asked to evaluate your study, which can be found at the end of this email.

As you will see, all three referees acknowledge the potential high interest of the findings. However, all three referees have raised a number of concerns and suggestions to improve the manuscript, or to strengthen the data and the conclusions drawn, which need to be addressed during the revision. As the reports are below, I will not detail them here. Given these constructive comments, we would like to invite you to revise your manuscript with the understanding that all referee concerns must be fully addressed in the revised manuscript and in a complete point-by-point response. Acceptance of your manuscript will depend on a positive outcome of a second round of review. It is EMBO reports policy to allow a single round of revision only and acceptance or rejection of the manuscript will therefore depend on the completeness of your responses included in the next, final version of the manuscript.

REFeree REPORTS

Referee #1:

The Manuscript by Meier T. et al. reports biochemical properties and cryoEM structure of N-type ATPase c-ring of *B. Pseudomallei*. They found that the c-ring is composed of 17 protomers which is the highest number in any types of rotary motors so far. The impact of the findings is probably

sufficient for publication in a high quality journal such as EMBO Reports. Further, the paper is well written and clear, and the information presented in the paper is important not only in rotary ATPase field but also in various other fields. Therefore, I strongly recommend this manuscript for publication in EMBO Reports.

Several changes should be made before the manuscript is published.

Some questions/remarks:

Results: Ion specificity

(1) The authors concluded that the *B. pseudomallei* c-ring is H⁺ specific, based on the biochemical findings using NCD-4. I agree that the main coupling ion of the c-ring is H⁺. However, I'm wondering that the c-ring can bind Na⁺ (Li⁺, Cs⁺??) at low concentration of H⁺ (i.e. alkali pH), because NCD-4 labeling is reduced by addition of Na⁺ even in pH 6.0. I think that the reduction of the labeling rate might suggest protective effect of Na⁺ on the DCCD reaction as described previous papers, rather than other effects that authors suggested in the manuscript. It would seem necessary to explain the possibilities and your opinion in detail, or to include additional data to prove it, such as pH dependence (pH 6.0-9.0) of the Na⁺ effect and other cations effect (e.g. K⁺, Cs⁺, Ca²⁺) as controls in the NCD-4 labeling experiments, if authors want to adhere to their decision.

Results; Regarding atpQ

(2) The authors mentioned the atpQ encodes a four TM helix protein of unknown function. Is there a possibility that q protein (AtpQ) is located in the c-ring like as that in a hybrid ring of *A. woodii* ATP synthase?

Minor comments:

Materials and Methods; NCD-4 modification part,

- (3) The value of lambda-emission (324 nm) seems to be wrong.
 (4) I think it would be better to show how to change the pH to 9.

Figure S4A,

- (5) Image resolution of the sequence alignments is low.

 Referee #2:

This manuscript investigates the structure and ion specificity of the novel N-type ATPase that is closely analogous to the F-type ATP synthase. Much less is understood about the N-type than the F-, A-, or V-type ATPases. The operon for the N-type is always found in addition to other F-/A-type atp-operons in the same genome, which apparently makes it redundant. The authors determine that the N-type ATPase from the Gram-negative, pathogenic beta-proteobacterium *Burkholderia pseudomallei* is a proton-dependent ATP synthase contrary to predictions from other research groups. Cryo-EM results of the c-ring rotor of the No subcomplex, which is responsible for generating torque from a transmembrane proton gradient, contains 17 c-subunits. Since each c-subunit in this complex is capable of translocating a proton, this structure represents the largest ion-to-ATP ratio identified to date. As such, this work represents a significant contribution to the understanding of these rotary ATP synthases. Two concerns must be addressed before the manuscript is acceptable for publication.

1. Based on the experiments of Figure 3, the authors conclude that the data indicate that the c17-ring is proton specific. However, the data do show that 15 mM NaCl significantly decreases the rate of NCD-4 labeling. In the text on p.7, the authors claim that this NaCl-dependent decrease is most likely due to the combined effects of salts on fluorescence quenching etc. They then state, "This observation contrasts with the strong and immediate effect of Na⁺ on NCD-4 labelling of Na⁺ binding c-rings (Meier et al., 2003)... The authors need to be more quantitative in the extent of inhibition observed in this manuscript relative that observed at the same NaCl concentration in Meier et al. 2003. What concentrations of NaCl are known Na⁺ dependent c-rings subjected to in vivo? How does that compare to the experimental data with the *B. pseudomallei* N-type?

2. At the bottom of p. 10, "We conclude that the structures of the neighboring subunits in the No subcomplex are overall conserved and similar to those of the conventional F-type ATPases. This appears to include in particular the lengths of helices 5 and 6 of the α -subunit..." This is speculative and should be moved to the Discussion. In particular, examination of the sequence comparisons provided in the supplementary information reveals some problems with this conclusion. For example, *E. coli* subunit- α residues E219 and H245 have been found to be adjacent to each other in various structures, even when their positions are swapped in some organisms. These residues have also been found to be of functional significance in the F-type. However, in the No subunit- α , neither residue appears to be conserved in the sequence alignment shown.

Referee #3:

Molecular architecture of a novel N type ATPase rotor ring from *Burkholderia pseudomallei* determined by electron cryo microscopy

1. Does this manuscript report a single key finding?

YES. Well, two: the *Burkholderia pseudomallei* N-type ATPase is proposed to be exclusively proton-coupled and has a uniquely high C17 c-ring symmetry.

2. Is the reported work of significance (YES), or does it describe a confirmatory finding or one that has already been documented using other methods or in other organisms etc (NO)?

YES

3. Is it of general interest to the molecular biology community?

YES. Rotary ATPases are a ubiquitous piece of cellular machinery, This work extends our knowledge of the poorly characterised N-type sub-family and so enriches our broader understanding of rotary ATPases in general.

4. Is the single major finding robustly documented using independent lines of experimental evidence (YES), or is it really just a preliminary report requiring significant further data to become convincing, and thus more suited to a longer" format article (NO)?

Not sure about this. Both the structural work (resolution too low, $6 \approx$ map interpreted at the residue level) and the biochemical data (NCD-4 reaction data, gold-labelling of vesicle reconstitution) feel somewhat preliminary. The authors probably did not want to spend too much time when the amount of novelty to be deduced is probably low and will remain low even when more and more definitive data can be obtained. The amount of sequence conservation to known systems, however, may well make this approach safe enough.

1. An initial paragraph that summarises the major finding and the referee's overall impressions, as well as highlighting any major strengths or shortcomings of the manuscript.

The authors carry out the first structural characterisation of an N-type ATPase c-ring, from *Burkholderia pseudomallei*. They reveal a 17-membered ring, thus increasing the highest known rotary ATPase c-ring subunit stoichiometry from 15. They also demonstrate biochemically that the *B. pseudomallei* N-type ATPase is probably exclusively proton coupled, an interesting finding given that the only other characterised N-type ATPase is Na^+ - coupled. The discussion concludes that the investigated N-type ATPase could be a proton pump and this has important implications for pathogenicity of the organism.

The findings are reasonably well supported by experimental evidence, however explanation of the data could be improved, especially the biochemical data. The resolution of the cryoEM map could be improved and this would make the work much more definitive.

Individual comments in no particular order:

1. 'novel (N-type) rotary ATPase' - the word novel should no be used in the scientific literature.

2. The abstract seems to overstate differences. Later it becomes clear that the c-ring is very similar to the one from F-ATPases and this finding actually strengthens the study as it makes the residue-level interpretation possible.

3. In the below segment, the actual work carried out should be explained more clearly. The N1NO ATPase was not expressed heterologously, rather 'quasi-homologously' (as the authors later state) and this work was separate from the cryoEM and biochemical studies of the heterologously expressed isolated c-ring:

"We studied a heterologously expressed *B. pseudomallei* N1No-type ATPase and investigated the structure and ion specificity of its membrane- embedded c-ring rotor by single-particle electron cryo-microscopy."

4. It is not clear why the various methods for solubilising the c-ring complex are discussed in the abstract.

5. Discussion of c-ring stoichiometry should make clear the distinction between subunit stoichiometry, number of ion-binding sites and the number of transmembrane (TM) helices. Perhaps noting the 4 TM helix (double hairpin) c subunit, bearing a single ion-binding site, of V-type ATPases here would help readers understand the relative importance of the 17-mer stoichiometry identified in this work.

6. Expression of the Bp N-type ATPase and the purification of the c-ring: moving paragraph 2 so it comes after paragraph 4 would make the distinction between expression of the entire N1No ATPase and the isolated c-ring much clearer.

7. In paragraphs 3 and 4 the use of the verb 'prove' is not warranted. Experiments cannot prove anything, it is a theoretical / mathematical (/ religious) concept. (Sorry for the rant).

8. Section 3.2 ion specificity: this section would benefit from a clearer explanation of what observed results would indicate H⁺/Na⁺ specificity, and the molecular explanation for these results. As it stands, this section is not convincing to the uninitiated reader. Could NCD-4 somehow react differently with this particular protein? Why was this possibility excluded? How would one exclude it? Admittedly, this reviewer is not an expert in this area and not familiar with this particular experiment.

9. Section 3.3, single particle cryoEM. The first paragraph is a good summary of the methodological difficulties associated with solubilising the c-ring oligomer. However the continuing discussion in paragraphs 2 and 3 about the choice of surfactant does not add to the biological message of the work, and is unlikely to be of interest to anyone except those trying to solubilise membrane complexes. While recording this sort of discussion in the literature is useful for future workers, almost all of this section could be moved to a supplementary discussion/methods and significantly streamline the report in the process.

10. Resolution of the resulting map. In 2016 I think it is disappointing to see a cryoEM manuscript with an atomic model refined into a $6 \approx$ map. Was it really not possible to push this further? The number of particles is not huge, given the admittedly small size of the complex (but because it is hollow, it should work a bit like a bigger protein). How was the model refined in Phenix? Real space refinement? Were any constraints used to keep secondary structure intact? Discussing the atomic model in such detail is somewhat alarming, although it is clear that the structure is very similar to what we know already, so most if it is probably right.

11. Section 3.4, model building and structure analysis: the 4th paragraph, titled 'No subunits in contact with the c-ring' is not structure analysis, it is sequence analysis and is not hugely convincing. In particular the possible role of AtpQ, which is presumably located in the membrane in the vicinity of the non-c subunits, is not discussed. If this paragraph is retained it should be moved to a sequence analysis section of the results, or placed in the discussion in a shortened form.

12. Discussion: in paragraph 2 the question of c-subunit stoichiometry/ring helix number/ion-

binding sites is discussed, in a much clearer way than in the introduction. The report would be improved if some of this clarifying information were added to the introduction.

13. Discussion paragraph 3 reports an interesting speculation on the physiological role of the N-ATPase during the early stages of infection, and also in cell homeostasis. The proposed role in homeostasis is not clear and needs further explanation.

14. Materials and methods are well and thoroughly explained.

15. 'We adapted our usual procedure ...' is a bit casual, sorry. What is the usual procedure, briefly?

16. 'This indicates that the *B. pseudomallei* c-oligomer does not alter its stoichiometry' - is it really possible to be sure about this with the method employed, even +/- 1 subunit since no control experiment can easily be performed.

17. '... labelling of Na⁺ binding c-rings (Meier et al., 2003)' - what c-rings were those from (what complexes/organisms)?

18. Different numbers of particles went into the classes for different detergents, so these cannot really be compared?

19. Was it tried to combine all particles, from all detergents?

20. Can the idea that it is a pump, needed for pathogenesis be tested (not suggesting the experiment, just asking).

21. Panel 2C is of low quality. Why was this done in negative stain? Cryo not possible?

1st Revision - authors' response

20 December 2016

Referee #1:

The Manuscript by Meier T. et al. reports biochemical properties and cryoEM structure of N-type ATPase c-ring of *B. Pseudomallei*. They found that the c-ring is composed of 17 protomers which is the highest number in any types of rotary motors so far. The impact of the findings is probably sufficient for publication in a high quality journal such as EMBO Reports. Further, the paper is well written and clear, and the information presented in the paper is important not only in rotary ATPase field but also in various other fields. Therefore, I strongly recommend this manuscript for publication in EMBO Reports.

RESPONSE: We thank the reviewer for his/her overall positive recommendation.

Several changes should be made before the manuscript is published. Some questions/remarks:

Results: Ion specificity

(1) The authors concluded that the *B. pseudomallei* c-ring is H⁺ specific, based on the biochemical findings using NCD-4. I agree that the main coupling ion of the c-ring is H⁺. However, I'm wondering that the c-ring can bind Na⁺ (Li⁺, Cs⁺??) at low concentration of H⁺ (i.e. alkali pH), because NCD-4 labeling is reduced by addition of Na⁺ even in pH 6.0.

RESPONSE: At alkaline pH, the carboxylates of the c-ring ion binding sites are deprotonated, so NCD-4 labelling does not occur (Figure 3B). We are therefore unable to provide a definitive answer to reviewer's question. It may be possible for the c-ring to bind Na⁺ (or Li⁺) at high pH if the proton concentration falls under a certain threshold level. This depends on the actual architecture of the c-ring ion-binding site. Based on our detailed knowledge of other c-rings, we suggest that the N-type c-ring is unable to bind Na⁺ (Li⁺, Cs⁺) even at high pH (e.g. pH 9.0), because so far, we

have not seen any structure of a c-ring, which would switch from H^+ binding at physiological (low) pH to Na^+ binding at pH 9 or 10. We cannot answer the reviewer's question until the structure of this c-ring is solved at high pH, in the presence of Na^+ (or Li^+ or Cs^+) and to a resolution at which the exact coordination geometry of the substrate ion becomes visible. This is clearly far beyond the scope of the present study. We prefer not to speculate on this point in the manuscript.

I think that the reduction of the labeling rate might suggest protective effect of Na^+ on the DCCD reaction as described previous papers, rather than other effects that authors suggested in the manuscript. It would seem necessary to explain the possibilities and your opinion in detail, or to include additional data to prove it, such as pH dependence (pH 6.0-9.0) of the Na^+ effect and other cations effect (e.g. K^+ , Cs^+ , Ca^{2+}) as controls in the NCD-4 labeling experiments, if authors want to adhere to their decision.

RESONSE: We agree that the ion specificity in this case is not as clear-cut as in other examples we have published, e.g. the c-ring from *Ilyobacter tartaricus* (exclusively Na^+ dependent, Meier et al., 2003) or the c-ring from *Bacillus pseudofirmus* OF4 (exclusively H^+ dependent, Preiss et al., 2010). It is therefore understandable that our new data raise this question. Our results actually reflect the considerable difficulty in collecting such data and the limitations of this type of experiment. Given this limitation, we state in the revised manuscript that the *B. pseudomallei* c-ring predominantly binds H^+ , in line with the reviewer's perception. At the same time, we extended our reasoning about the factors that can influence our measurement presented in Figures 3 and S2 in the results section.

The whole results section has been completely revised and extended with respect to the NCD-4 labeling, including also a more quantitative assessment with a new Table S1, as requested by reviewer 2, and a more detailed explanation of the experiments, as requested by reviewer 3. We will, however, not include a pH dependence study or test more cations because this would not change any of the conclusions that we can draw from the experiments at this stage. We are convinced that the relevant cations (Na^+ , Li^+ and Cs^+) for the question at hand are sufficiently investigated. The binding of K^+ (larger radius than Na^+ , already covered by Cs^+) or even bivalent Ca^{2+} to an ATP synthase would be a purely chemical question and does not make sense from a physiological point of view. This is the revised NCD-4 section:

Ion specificity. To address the ion specificity of the isolated *B. pseudomallei* c-ring oligomer, the detergent-solubilized complex was exposed to *N*-cyclohexyl-*N'*-(4-(dimethylamino)- α -naphthyl)-carbodiimide (NCD-4), a fluorescent analogue of the ATP synthase inhibitor *N,N'*-dicyclohexylcarbodiimide (DCCD) (**Fig 3** and **Fig. S2**). NCD-4 reacts covalently with the protonated, ion-binding carboxylates of c-rings. Therefore it can be used to directly characterize the type of ion that is bound in a given ATP synthase by measuring a continuous increase of fluorescence upon addition of the fluorophore. For example, in the case of the Na^+ binding c-ring from *I. tartaricus*, the reaction can be specifically and immediately stopped by adding 15 mM NaCl resulting in 1% remaining labeling activity (Meier et al., 2003). Similarly, in a H^+ binding c-ring, the increase of fluorescence is completely stopped (no measurable remaining activity) by deprotonation of the ion binding carboxylates, caused by the shift of pH to 9.0 in the cuvette (Preiss et al., 2010). For testing our *B. pseudomallei* c-oligomer, we chose to initiate the reaction at pH 6.0 at which the ion binding c-oligomer carboxylates are still protonated and is able to react rapidly with NCD-4, while all water-exposed carboxylates (e.g. loop region, **Figure 1B**) are deprotonated and are not involved in the reaction. A continuous increase of fluorescence was monitored for several minutes and then quantified. The reaction starts with a linear increase of fluorescence (**Fig. 3A**, blue arrow). We then tested the effect of Na^+ (**Fig. 3A**, red arrow) and depletion of H^+ (pH 9.0) on the reaction efficiency (**Fig. 3B**, red arrow). Li^+ and Cs^+ ions were used as controls to test the effect of other, non-physiological monovalent cations on the labeling reaction (**Fig. S2**). The addition of 15 mM Na^+ reduced NCD-4 labeling from the initial rate (defined as 100%) to 40% remaining labeling activity. The effect of 15 mM Li^+ (55% remaining) and Cs^+ (44% remaining) was similar, while a pH increase to 9.0 (=complete deprotonation of c-ring glutamates) again resulted in an immediate and complete stop of the reaction with only 1% of rest labeling. A further increase of the Na^+ , Li^+ and Cs^+ concentration (150 mM) to a value similar to a physiological concentration of blood serum decreased the total NCD-4 labeling efficiency to 27%, 17% and 24%, respectively (**Fig. 3A** and **Fig. S2**). A quantification of the labeling efficiencies is given in **Table S1**. Note that the rather high increase of cationic strength in the measurement causes effects on fluorescence quenching of the fluorescent probe, as it has been described previously (Adenier & Aaron, 2002). In

addition, the combined effects of salt on the availability of water and the critical micelle concentration of the detergent need to be taken into consideration. Importantly however, the observation contrasts with the strong and immediate effect of Na^+ on NCD-4 labeling of Na^+ binding c-rings from *I. tartaricus* (Meier et al., 2003), while it matches that observed with a c-ring isolated from a H^+ ATP synthase (Preiss, Yildiz et al., 2010) remarkably well. In contrast to what had been predicted from the polypeptide sequence (Dibrova et al., 2010), these biochemical data suggest (SKIPPED: show unequivocally) that the *B. pseudomallei* N-type ATPase c-ring is predominantly H^+ selective, indicating it is coupled to protons and hence contributes to or utilizes the proton gradient, not the sodium gradient, across the bacterial membrane.

Results; Regarding atpQ

(2) The authors mentioned the atpQ encodes a four TM helix protein of unknown function. Is there a possibility that q protein (AtpQ) is located in the c-ring like as that in a hybrid ring of *A. woodii* ATP synthase?

RESPONSE: As far as we can tell, there is no protein in the pore of the A. woodii c-ring (Matthies et al., Nat. Comm. 2014). What the reviewer may have in mind is the N-terminal region of the A. woodii c₁ subunit, which stretches horizontally across the c-ring on the periplasmic side. To test the reviewer's proposal we aligned the amino acid sequence of the c₁ N-terminus and compared it to the putative membrane protein q. We found that there is no significant conservation between the two sequences. Along the reviewer's suggestion, we further considered the possibility that the atpQ gene product could perhaps resemble the recently described protein YPR170W-B, which was found to be part of the inner c-ring pore in a yeast V-type ATPase (Mazhab-Jafari et al., Nature 2016). Unfortunately also in this case, we were unable to find any similarity between these two proteins. At this stage, any statement about the function or location of protein q would be purely speculative.

Minor comments:

Materials and Methods; NCD-4 modification part,

(3) The value of lambda-emission (324 nm) seems to be wrong.

RESPONSE: Thank you for alerting us to this mistake. We corrected: $\lambda_{em} = 440 \text{ nm}$

(4) I think it would be better to show how to change the pH to 9.

RESPONSE: We added this information in the Figure 3 legend: "(B) Increasing the pH to 9.0 by adding 2.5 M Tris base stock solution (red arrow) abolished..." We now describe the blue and red arrows in the figure legend.

Figure S4A,

(5) Image resolution of the sequence alignments is low.

RESPONSE: We improved the resolution

Referee #2:

This manuscript investigates the structure and ion specificity of the novel N-type ATPase that is closely analogous to the F-type ATP synthase. Much less is understood about the N-type than the F-, A-, or V-type ATPases. The operon for the N-type is always found in addition to other F-/A-type atp-operons in the same genome, which apparently makes it redundant. The authors determine that the N-type ATPase from the Gram-negative, pathogenic beta-proteobacterium *Burkholderia pseudomallei* is a proton-dependent ATP synthase contrary to predictions from other research groups. Cryo-EM results of the c-ring rotor of the No subcomplex, which is responsible for generating torque from a transmembrane proton gradient, contains 17 c-subunits. Since each c-subunit in this complex is capable of translocating a proton, this structure represents the largest ion-to-ATP ratio identified to date. As such, this work represents a significant contribution to the understanding of these rotary ATP synthases.

RESPONSE: We thank the reviewer for his/her positive evaluation of our work.

Two concerns must be addressed before the manuscript is acceptable for publication.

1. Based on the experiments of Figure 3, the authors conclude that the data indicate that the c17-ring is proton specific. However, the data do show that 15 mM NaCl significantly decreases the rate of NCD-4 labeling. In the text on p.7, the authors claim that this NaCl-dependent decrease is most likely due to the combined effects of salts on fluorescence quenching etc.

*RESPONSE: See our response to reviewer 1 above. We revised all passages of the manuscript stating that H^+ exclusively binds to this c-ring and changed it in the way that the *B. pseudomallei* c-ring preferentially binds H^+ . We furthermore extended our discussion of the side effects of dilution in the cuvette during the experiment and the high ion concentrations that would quench the fluorophore. We added the following reference to support this statement: "Adenier A & Aaron JJ (2002) A spectroscopic study of the fluorescence quenching interactions between biomedically important salts and the fluorescent probe merocyanine 540. Spectrochim Acta A Mol Biomol Spectrosc 58(3):543-551."*

They then state, "This observation contrasts with the strong and immediate effect of Na^+ on NCD-4 labelling of Na^+ binding c-rings (Meier et al., 2003)... The authors need to be more quantitative in the extent of inhibition observed in this manuscript relative that observed at the same NaCl concentration in Meier et al. 2003.

*RESPONSE: in this manuscript and compared it to a previous observation with a Na^+ binding c-ring: "[...] For example, in the case of the Na^+ binding c-ring from *I. tartaricus*, the reaction can be specifically and immediately stopped by adding 15 mM NaCl, resulting in 1% remaining labeling activity (Meier et al., 2003) [...]"*

The results part now contains numerous quantitative measurements. We also added a new Table S1 to the Supplementary materials, which summarizes the labeling quantifications:

New Table S1. Quantification of NCD-4 labeling efficiency of the *B. pseudofirmus* c17 ring

	NaCl, pH 6 (mM)			LiCl, pH 6 (mM)			CsCl, pH 6 (mM)			pH 9 shift	
	0	15	150	0	15	150	0	15	150	pH 6	pH 9
labeling*	100%	40%	27%	100%	55%	17%	100%	44%	24%	100%	1%
Time (min)	0	6.5	13	0	6.5	13	0	6.5	13	0	(6.5)**

*the initial labeling rate was taken as 100%.

**immediately after addition

What concentrations of NaCl are known Na^+ dependent c-rings subjected to in vivo?

RESPONSE: The c-ring in vivo is almost completely embedded in the hydrophobic lipid bilayer, where it is in contact with various other ATPase subunits. Here are four examples of species that have Na^+ -dependent ATP synthases. The c-rings face these Na^+ concentrations on the periplasmic side:

- Ilyobacter tartaricus and Propionigenium modestum: These bacteria were originally isolated from channel sediments in Venice, Italy. The Na^+ concentration of sea water is approximately 0.6-1 M. To isolate the ATP synthase, *I. tartaricus* is grown in medium that contains 20 g/l NaCl (around 330 mM Na^+).*
- Fusobacterium nucleatum: This opportunistic pathogen is typically found in human dental plaque. The Na^+ content of saliva is around 2-21 mM. *F. nucleatum* can also infect organs, e.g. lung, liver and brain. The Na^+ concentrations in these organs is around 150 mM (estimated). Human blood serum is around 144 mM.*

- *Acetobacterium woodii* was originally isolated from a marine estuary (Na^+ concentration varies) and it is typically grown in medium that contains about 18 mM NaCl.

On the cytoplasmic side of the c-ring, typically, the inside of a cell (cytosol) has low Na^+ , for example 12 mM. How does that compare to the experimental data with the *B. pseudomallei* N-type?

*RESPONSE: Our experimental data show Na^+ concentrations tested at 15 mM (a typical cytosolic concentration) and 150 mM (the typical concentration of human blood), which well reflect the in vivo situation of *B. pseudomallei*. In response to this reviewer's question, we added: "A further increase of the Na^+ , Li^+ and Cs^+ concentration (150 mM) to a value similar to a physiological concentration of blood serum decreased the NCD-4 labeling efficiency to 27%, 17% and 24%, respectively."*

2. At the bottom of p. 10, "We conclude that the structures of the neighboring subunits in the No subcomplex are overall conserved and similar to those of the conventional F-type ATPases. This appears to include in particular the lengths of helices 5 and 6 of the a-subunit..." This is speculative and should be moved to the Discussion. In particular, examination of the sequence comparisons provided in the supplementary information reveals some problems with this conclusion. For example, *E. coli* subunit-a residues E219 and H245 have been found to be adjacent to each other in various structures, even when their positions are swapped in some organisms. These residues have also been found to be of functional significance in the F-type. However, in the No subunit-a, neither residue appears to be conserved in the sequence alignment shown.

*RESPONSE: We agree that there is biochemical evidence that E219 can be swapped with H245 in *E. coli* and that they are close to each other in the structure. However, neither the individual residues nor the pair is highly conserved in different species. Would the reviewer expect conservation in the N-type sequences whereas other species do not show a corresponding conservation in the F-type sequences? This point becomes clearer with a closer look at some of these pairs in bacterial ATP synthases (we have chosen examples from bacterial F-type ATPases) and the N-type ATPase from *B. pseudomallei*:*

E. coli :	Glu219	His245
I. tartaricus :	Met	Asp
M. tuberculosis :	Gly	Asp
T. maritima :	Ala	Gly
W. succinogenes :	Asp	Gly
B. pseudomallei :	Val	Asp

*The list shows that the amino acids themselves are not conserved in bacteria but the pair shows some degree of conservation (but not in all cases, e.g. *T. maritima*) with respect to a negatively charged residue (either Glu or Asp) on the one residue and a hydrophobic (or no, Gly) side chain on the other residue of the pair. Notably, *T. maritima* seems the odd one out (there are more such bacterial sequences), but the degree of conservation in this pair is also found in the case of the *B. pseudomallei* N-type a-subunit: it has a pair consisting of a hydrophobic Valine and a negatively charged Aspartate.*

- ➔ *The reviewer's point that E219 and H245 is conserved (or conserved in swapped orientation) is not valid. We however agree that there is some degree of conservation with respect to the negative charge in pair with a hydrophobic residue.*
- ➔ **B. pseudomallei* actually has this conserved pair in the N-type subunit sequence. So there is no problem from this side. Besides that, the *B. pseudomallei* sequence also shows many other a-subunit typical features, such as the high conservation of helices 5 and 6, or the overall conserved and functionally important arginine (R210 in *E. coli*) embedded in the sequence RLF_{GN} of helix 5. This is all indicated in our alignment shown in Figure S4.*
- ➔ *There can be no doubt that this sequence is an a-subunit homologue sequence.*

*For these reasons the reviewer's point is not justified for other a-subunit sequences than *E. coli*; we stayed with our conclusions and consider it as a result of our investigation, so we did not move this point to the discussion.*

 Referee #3:

1. Does this manuscript report a single key finding?

YES. Well, two: the Burkholderia pseudomallei N-type ATPase is proposed to be exclusively proton-coupled and has a uniquely high C17 c-ring symmetry.

2. Is the reported work of significance (YES), or does it describe a confirmatory finding or one that has already been documented using other methods or in other organisms etc (NO)?

YES

3. Is it of general interest to the molecular biology community?

YES. Rotary ATPases are a ubiquitous piece of cellular machinery, This work extends our knowledge of the poorly characterised N-type sub-family and so enriches our broader understanding of rotary ATPases in general.

4. Is the single major finding robustly documented using independent lines of experimental evidence (YES), or is it really just a preliminary report requiring significant further data to become convincing, and thus more suited to a longer format article (NO)?

Not sure about this. Both the structural work (resolution too low, 6 Å map interpreted at the residue level) and the biochemical data (NCD-4 reaction data, gold-labelling of vesicle reconstitution) feel somewhat preliminary. The authors probably did not want to spend too much time when the amount of novelty to be deduced is probably low and will remain low even when more and more definitive data can be obtained. The amount of sequence conservation to known systems, however, may well make this approach safe enough.

1. An initial paragraph that summarises the major finding and the referee's overall impressions, as well as highlighting any major strengths or shortcomings of the manuscript.

The authors carry out the first structural characterisation of an N-type ATPase c-ring, from Burkholderia pseudomallei. They reveal a 17-membered ring, thus increasing the highest known rotary ATPase c-ring subunit stoichiometry from 15. They also demonstrate biochemically that the B. pseudomallei N-type ATPase is probably exclusively proton coupled, an interesting finding given that the only other characterised N-type ATPase is Na⁺-coupled. The discussion concludes that the investigated N-type ATPase could be a proton pump and this has important implications for pathogenicity of the organism. The findings are reasonably well supported by experimental evidence, however explanation of the data could be improved, especially the biochemical data. The resolution of the cryoEM map could be improved and this would make the work much more definitive.

RESPONSE: We appreciate the reviewer's overall positive attitude towards our work and thank him/her also for the critical comments.

Individual comments in no particular order:

1. 'novel (N-type) rotary ATPase' - the word novel should no be used in the scientific literature.

RESPONSE: We followed the nomenclature used in Armen Mulkidjanian's paper (Dibrova et al., 2010) in which the N-type ATPase was introduced as a "novel" type of ATPase subfamily always present "next to" an F-type ATPase. Due to these two characteristics (novel and next to) and its predicted "Na⁺ dependency" (which B. pseudomallei apparently does not share) the enzyme was named N-type. To keep this connection with its origin we follow this nomenclature, but we slightly modified the introduction to clarify this point better: "The protein complexes encoded by these so-called "novel" or "next to" atp-operons are referred to as N-type ATPase and were predicted to be "predominantly Na⁺-selective" (Dibrova et al., 2010)."

2. The abstract seems to overstate differences. Later it becomes clear that the c-ring is very similar to the one from F-ATPases and this finding actually strengthens the study as it makes the residue-level interpretation possible.

RESPONSE: The abstract states that the N-ATPase is a novel type of enzyme and that its structure and function is unexplored. After our structure determination and biochemical work we can conclude that the N-type rotor ring is very similar to F-type rotor rings at the level of overall architecture and proton-binding properties of individual c-subunits, but it is different and so far unique with respect to its stoichiometry (which has important functional implications, as presented). This is the first contribution of structural work to an N-type ATPase and its functional rationalization. We cannot follow the reviewer's view that the abstract is overstated.

3. In the below segment, the actual work carried out should be explained more clearly. The N1N0 ATPase was not expressed heterologously, rather 'quasi-homologously' (as the authors later state) and this work was separate from the cryoEM and biochemical studies of the heterologously expressed isolated c-ring: "We studied a heterologously expressed *B. pseudomallei* N1N0-type ATPase and investigated the structure and ion specificity of its membrane-embedded c-ring rotor by single-particle electron cryo-microscopy."

RESPONSE: We removed "heterologously" from the abstract, as it unnecessarily adds weight. In the manuscript (results and methods) we then explain the details how exactly the material was produced.

4. It is not clear why the various methods for solubilising the c-ring complex are discussed in the abstract.

RESPONSE: We modified the sentence to emphasize better why the choice of detergent was important for the quality of the structural data collected: "Of several amphiphilic compounds tested for solubilizing the complex, the choice of the low-density, low-CMC detergent LDAO was optimal in terms of map quality and resolution."

5. Discussion of c-ring stoichiometry should make clear the distinction between subunit stoichiometry, number of ion-binding sites and the number of transmembrane (TM) helices. Perhaps noting the 4 TM helix (double hairpin) c subunit, bearing a single ion-binding site, of V-type ATPases here would help readers understand the relative importance of the 17-mer stoichiometry identified in this work.

*ANSWER: It may have escaped the reviewer's attention that we had already dedicated a complete paragraph to exactly this point in the discussion: "Each c-subunit in the *B. pseudomallei* c₁₇ ring is a hairpin of two α -helices with an ion-binding site at the interface between adjacent c-subunits, as in the F-type ATPases. The K-rings of the V-type ATPases are even larger, equivalent to 20 single-hairpin c-subunits, but these larger rings only contain 10 ion-binding sites, owing to the fact that each K-subunit is a double helix hairpin with only one single acidic ion-binding glutamate (Murata et al., 2005). The high degree of sequence similarity between N-type and F-type operons supports the notion that an N₁N₀-type ATPase shares the same structural features and functionalities of an F₁F₀-ATPase. However, given that the *B. pseudomallei* N-type ATPase can translocate 17 ions per complete rotation, this results in an unusually high ion-to-ATP ratio of 5.7."*

6. Expression of the Bp N-type ATPase and the purification of the c-ring: moving paragraph 2 so it comes after paragraph 4 would make the distinction between expression of the entire N1N0 ATPase and the isolated c-ring much clearer.

ANSWER: We agree and followed this suggestion in the revised manuscript.

7. In paragraphs 3 and 4 the use of the verb 'prove' is not warranted. Experiments cannot prove anything, it is a theoretical / mathematical (/ religious) concept. (Sorry for the rant).

ANSWER: We replaced "prove" by "show."

8. Section 3.2 ion specificity: this section would benefit from a clearer explanation of what observed results would indicate H⁺/Na⁺ specificity, and the molecular explanation for these results. As it stands, this section is not convincing to the uninitiated reader.

ANSWER: To follow this proposal and initiate the reader better we have carefully revised and extended the ion specificity section quite extensively. We introduce the usage of NCD-4 and its capability to react with protonated carboxylates, hence its usability to track Na⁺ binding in c-rings. We also explain with referenced examples what happens if the c-ring faces Na⁺ or high pH. In this context and in response to Reviewer 1 we have modified our statement that the c-ring exclusively binds Na⁺ and we now state that this ATPase preferentially binds protons. It does not change the overall perception that this N-type ATPase uses protons, but it better reflects the limitations of the experiment (see response to reviewer 1 above). Finally we have added more quantitative data, as requested by reviewer 2, and added a new Supplementary Table S1 to support our statements and compare the data with previously made NCD-4 experiments with Na⁺ and H⁺ binding c-rings.

Could NCD-4 somehow react differently with this particular protein? Why was this possibility excluded? How would one exclude it? Admittedly, this reviewer is not an expert in this area and not familiar with this particular experiment.

ANSWER: Indeed NCD-4 can have unspecific side reactions. For example, NCD-4 can also react with all protonated carboxylates in the protein (the c-subunit contains a total of 4 Glu or Asp, in particular at the solvent exposed sites (e.g. two Asp at the cytoplasmic side of the c-ring, see Figure 1B). However at the measured pH (6.0), the water-exposed carboxylates are deprotonated (the pKa of such a carboxylate is approximately 4.5), hence the side reactions at the measured pH are negligible. The chosen pH (6.0) represents the best possible choice between “too many side reactions” and “labelling too slow”. This is the result of a biochemical experiment which (as often) is not black and white (see also response to reviewer 1 above).

9. Section 3.3, single particle cryoEM. The first paragraph is a good summary of the methodological difficulties associated with solubilising the c-ring oligomer. However the continuing discussion in paragraphs 2 and 3 about the choice of surfactant does not add to the biological message of the work, and is unlikely to be of interest to anyone except those trying to solubilise membrane complexes. While recording this sort of discussion in the literature is useful for future workers, almost all of this section could be moved to a supplementary discussion/methods and significantly streamline the report in the process.

ANSWER: We respectfully but strongly disagree with the reviewer’s opinion. Section 3.3 is an important part of this work. As we state, the choice of the detergent used was crucial for the quality of the EM data and our findings are thus important for the increasing crowd of people working with samples at the electron microscope.

We took datasets of the rotor ring in detergents that differ in their density and micelle size. The results clearly show that the lower the density and the smaller the micelle size (Table S1), the more details and important features of the c-ring become visible. This contribution will therefore not only be valuable from the point of view of the biological question addressed, but also for the structural biology community, which currently undergoes dramatic changes by the technical progresses made in electron microscopy. Our work shows how important the choice of the detergent or surfactant is for the final quality of the map of membrane protein complexes.

It may be true that this section is mainly of interest to those who solubilize membrane protein complexes for structure determination by cryoEM. However, the number of scientists who do this is increasing rapidly, as is the significance cryoEM for membrane protein structure determination. We therefore intend to keep this section in the main text but we emphasized the use of the detergents better in the abstract (Point 4, same reviewer).

10. Resolution of the resulting map. In 2016 I think it is disappointing to see a cryoEM manuscript with an atomic model refined into a 6 Å map.

RESPONSE: The reviewer may not be aware that image contrast, and hence map resolution, in cryoEM depends strongly on molecular mass, and, in the case of membrane proteins, on how much

of the protein protrudes into the aqueous medium (where contrast is higher). A 6.1 Å resolution map of such a small membrane protein that hardly protrudes from the lipid bilayer is actually quite remarkable. To our knowledge, this is the highest resolution to date achieved by cryoEM with a membrane protein of this kind. In this context, the choice of detergent (see point 9 above) is particularly important, as it optimizes the image contrast and hence enables high resolution.

Was it really not possible to push this further? The number of particles is not huge, given the admittedly small size of the complex (but because it is hollow, it should work a bit like a bigger protein).

RESPONSE: The resolution can always be pushed further. We estimate that with the current generation of direct electron detectors, it should be possible to improve the map resolution to the point where sidechains become visible (around 4 Å) by adding another 1.4 million particles of LDAO-solubilized N-type c-rings. This would require a massive amount of EM time that is, at present, simply not available for this project (which competes with numerous other cryoEM projects at the MPI of Biophysics). Although we will of course attempt to improve the resolution in the future, this would not make much difference for the biological questions that interest us, which can be answered very well at the present resolution. The main points here is the large size, highly unusual symmetry and the ion-binding specificity of this particular, novel c-ring rotor.

How was the model refined in Phenix? Real space refinement?

ANSWER: This was stated in the Methods section: "Seventeen copies of a helix hairpin [...] were fitted to the EM map and refined against the cryoEM density using real-space-refinement in Phenix (Fig. 4F)."

Were any constraints used to keep secondary structure intact?

ANSWER: No.

Discussing the atomic model in such detail is somewhat alarming, although it is clear that the structure is very similar to what we know already, so most if it is probably right.

ANSWER: We agree that our interpretation is accurate on the basis of previous knowledge that has been carefully incorporated into our data analysis. Nevertheless, the manuscript avoids the expression "atomic model" throughout, in contrast to many others in the current literature that use this term indiscriminately at this, or lower, resolution.

11. Section 3.4, model building and structure analysis: the 4th paragraph, titled 'No subunits in contact with the c-ring' is not structure analysis, it is sequence analysis and is not hugely convincing. In particular the possible role of AtpQ, which is presumably located in the membrane in the vicinity of the non-c subunits, is not discussed. If this paragraph is retained it should be moved to a sequence analysis section of the results, or placed in the discussion in a shortened form.

RESPONSE: The reviewer correctly states that the section about the c-ring neighbouring subunits is largely based on sequence analysis. We therefore clarify this in the subtitle: "Analysis of N_o subunits in contact with the c-ring based on sequence comparison." Nevertheless, this is a valid result that provides useful hints for the structural analysis of the c-ring neighbourhood. With respect to atpQ: We prefer not to speculate - please see our response to Reviewer 1 above. We have no functional or structural data about atpQ, so its role or possible location remains to be investigated in the future.

12. Discussion: in paragraph 2 the question of c-subunit stoichiometry/ring helix number/ion-binding sites is discussed, in a much clearer way than in the introduction. The report would be improved if some of this clarifying information were added to the introduction.

RESPONSE: Although we dedicated already a whole paragraph to the c-subunit and c-ring (and its ions) in the introduction we understand that the reviewer was missing the link between the number of subunits in a given c-ring and the resulting "ion-to-ATP" ratio thereof in the corresponding ATPase. We now added this missing link to the introduction: "[...] The c-ring stoichiometry defines

the number of ions to be transported along this pathway and hence the ion-to-ATP ratio in rotary ATPases/synthases of any given species [...]"

*The reader is informed that the c-subunit is key to determine the type and number of coupling ion in a given ATPase. This topic is later picked up and elaborated in the discussion section (see point 5 above, same Reviewer). We prefer to address the question of variability of c-ring stoichiometries after presenting the result that the *B. pseudomallei* c-ring has an unusually high stoichiometry.*

13. Discussion paragraph 3 reports an interesting speculation on the physiological role of the N-ATPase during the early stages of infection, and also in cell homeostasis. The proposed role in homeostasis is not clear and needs further explanation.

RESPONSE: We understand that the term "homeostasis" finally comes as a surprise in the last sentence of the Discussion. We therefore clarified the term homeostasis with respect to its meaning for the cell internal pH already earlier in the Discussion: "We propose that the N-type rotary ATPase, as a highly efficient ATP-driven proton pump, enables the bacterium to cope with acid stress and maintain its pH homeostasis to survive in this hostile environment, while the F-type ATP synthase generates ATP by oxidative phosphorylation (Fig. 5) [...]." The reader is also referred to Fig. 5. The figure 5 legend then explains all steps shown in the figure.

14. Materials and methods are well and thoroughly explained.

15. 'We adapted our usual procedure...' is a bit casual, sorry. What is the usual procedure, briefly?

RESPONSE: We clarified: "The c-subunit did not have an affinity tag, hence we used a purification method that bases on ammonium sulfate precipitation for the purification of c-rings (Meier et al., 2003)."

16. 'This indicates that the *B. pseudomallei* c-oligomer does not alter its stoichiometry' - is it really possible to be sure about this with the method employed, even +/- 1 subunit since no control experiment can easily be performed.

ANSWER: Yes, it is possible to be sure about this statement. Adding or removing one subunit would change the molecular weight of the c-ring by about 6%, which would become apparent on the SDS gel. We have investigated many such gel shifts in the past. Examples:

- c_{13} to c_{12} for the *B. pseudofirmus* OF4 c-ring: Preiss et al., PNAS 2013*
- c_{11} to c_{12} for the *I. tartaricus* c-ring: Pogoryelov et al., PNAS 2012*
- gel shifts of various cyanobacterial c-rings with c_{13} , c_{14} and c_{15} stoichiometries: Pogoryelov et al., 2008*
- Biochemical analysis of the causes of gel shifts observed in *I. tartaricus* and *P. modestum* c-rings (both c_{11} stoichiometries): Meier et al., FEBS J. 2005.*

17.... labelling of Na⁺ binding c-rings (Meier et al., 2003)' - what c- rings were those from (what complexes/organisms)?

*RESPONSE: We added: "... from *I. tartaricus*."*

18. Different numbers of particles went into the classes for different detergents, so these cannot really be compared?

RESPONSE: The quality of the classes that we obtained for the different detergents vary already at an earlier stage of data analysis, where fewer particles contribute. To make this point clearer, we have added a figure for the benefit of this reviewer, that shows the same data analysis with the same amount of particles (45000) in each case and compare it to the final result, as shown in the manuscript. The best classes were obtained again in LDAO, followed by DDM, C12E8 and Amphipol. We do not think the figure is should go in the manuscript, as it represents a preliminary stage of the data analysis.

19. Was it tried to combine all particles, from all detergents?

RESPONSE: Yes, we tried to combine the total number of particles (290'000) but the result was blurred by the lower quality of classes from the less optimal detergents.

20. Can the idea that it is a pump, needed for pathogenesis be tested (not suggesting the experiment, just asking)?

RESPONSE: Yes. It could for example be tested by knocking out the N-type ATPase in B. pseudomallei and determine and compare the pH of infected host cell phagosomes of wildtype and knock-out mice.

21. Panel 2C is of low quality. Why was this done in negative stain? Cryo not possible?

RESPONSE: Since our question could be already answered by negative stain, we saw no need for cryoEM (which is experimentally much more demanding) in this particular instance.

2nd Editorial Decision

11 January 2017

Thank you for the submission of your revised manuscript to our editorial offices. We have now received the reports from the three referees that were asked to re-evaluate your study that you will find enclosed below. As you will see, all three referees support the publication of your manuscript in EMBO reports. However, referee #3 has some further comments that we ask you to address in a final revised version. Further, I have several editorial requests.

For a Scientific Report we require that results and discussion be combined in one chapter called "Results and Discussion." Please do that for your manuscript.

Supplementary/additional data: The Expanded View format, which will be displayed in the main HTML of the paper in a collapsible format, has replaced the Supplementary information. You can submit up to 5 images as Expanded View. Please follow the nomenclature Figure EV1, Figure EV2 etc. The figure legend for these should be included in the main manuscript document file in a section called Expanded View Figure Legends after the main Figure Legends section. Additional Supplementary material should be supplied as a single pdf labeled Appendix. The Appendix includes a table of content on the first page, all figures and their legends. Please follow the nomenclature Appendix Figure Sx throughout the text and also label the figures according to this nomenclature. For more details please refer to our guide to authors. Thus, I would suggest to upload the supplement as separate EV figures (4) and EV tables (2). Please add the legends for these to the main manuscript text, after the main figure legends.

Supplemental Fig 4 (future EV4) takes up 2 pages, which is not in line with the requirements of our publisher. Please fit this on one page (or make two EV figures from this).

After you have finished with the EV figures and tables, please change and adjust the callouts in the manuscript text.

Important: All materials and methods should be included in the main manuscript file. Please move those from the supplement to the main text. Please also shorten the materials & methods part, if possible.

Please submit the movie and a text file with title and legend together in a zip file.

The title is currently too long. Please shorten it, or provide a different title with no more than 100 characters. The abstract is currently too long. Please shorten it to less than 176 words.

REFEREE REPORTS

Referee #1:

I don't see any need for further revisions/corrections. I'm sure that the manuscript is now suitable for publication in EMBO reports.

Referee #2:

The revised manuscript adequately addresses the concerns of all of the reviewers and should be accepted for publication.

Referee #3:

The authors have responded thoughtfully and constructively to the comments provided and I certainly recommend the much-improved manuscript for publication. A few comments and remaining points of concern are given below.

1. As the authors note in their rebuttal the use of the word 'novel' here is only appropriate in reference to the origin of the 'N' in N-type ATPase. The introduction is much improved by making this clear, however the title, abstract and key findings ('new (N-)type of ATPase rotor ring') remain misleading. The use of the word novel in reference to the c-ring only in the first line of the discussion is unnecessary and confusing.
2. Discussion of c-ring stoichiometry and ion-binding is improved. The terminology 'double helix hairpin' to refer to a 4 helix subunit is easy to parse incorrectly, although this is a question for the field.
3. Use of the word novel would (unusually) seem appropriate and improve clarity in para 3 of introduction 'An in silico study of the distribution of rotary ATPase operons in bacteria and archaea identified a NOVEL atp operon in several phylogenetically independent groups of bacteria and archaea (Dibrova, Galperin et al., 2010).
4. Discussion of ion specificity experiments is greatly improved and much more convincing as a result, mostly in response to reviewer 1 who did an excellent job at pointing out weaknesses and solutions. As the authors note the result seems to be that the pump is predominantly H⁺ specific, however in the final paragraph of the introduction it is still referred to as binding H⁺ exclusively. This should be changed as it has been done elsewhere.
5. I note the authors rationale for disagreement with my opinion that the discussion of detergent choice is inappropriate for the main text, and surely the abstract - I still disagree. In contrast, I still agree, as previously noted, that the experiments are important and interesting for workers in cryoEM of membrane proteins. Nevertheless I do think this section as it stands is suitable for publication and our disagreement reflects a matter of taste and style, mostly.

2nd Revision - authors' response

02 February 2017

Responses to Referee #3:

The authors have responded thoughtfully and constructively to the comments provided and I

certainly recommend the much-improved manuscript for publication. A few comments and remaining points of concern are given below.

1. As the authors note in their rebuttal the use of the word 'novel' here is only appropriate in reference to the origin of the 'N' in N-type ATPase. The introduction is much improved by making this clear, however the title, abstract and key findings ('new (N-)type of ATPase rotor ring') remain misleading. The use of the word novel in reference to the c-ring only in the first line of the discussion is unnecessary and confusing.

RESPONSE: We appreciate the rigorous revision of the reviewer and removed the word novel from the title and abstract. We ask the Editor also to remove novel also from sentence in the key findings as they have already been separately forwarded by the Editor.

2. Discussion of c-ring stoichiometry and ion-binding is improved. The terminology 'double helix hairpin' to refer to a 4 helix subunit is easy to parse incorrectly, although this is a question for the field.

3. Use of the word novel would (unusually) seem appropriate and improve clarity in para 3 of introduction 'An in silico study of the distribution of rotary ATPase operons in bacteria and archaea identified a NOVEL atp operon in several phylogenetically independent groups of bacteria and archaea (Dibrova, Galperin et al., 2010).

RESPONSE: We added novel here.

4. Discussion of ion specificity experiments is greatly improved and much more convincing as a result, mostly in response to reviewer 1 who did an excellent job at pointing out weaknesses and solutions. As the authors note the result seems to be that the pump is predominantly H⁺ specific, however in the final paragraph of the introduction it is still referred to as binding H⁺ exclusively. This should be changed as it has been done elsewhere.

RESPONSE: Thanks, we changed it.

5. I note the authors rationale for disagreement with my opinion that the discussion of detergent choice is inappropriate for the main text, and surely the abstract - I still disagree. In contrast, I still agree, as previously noted, that the experiments are important and interesting for workers in cryoEM of membrane proteins. Nevertheless I do think this section as it stands is suitable for publication and our disagreement reflects a matter of taste and style, mostly.

3rd Editorial Decision

09 February 2017

I am very pleased to accept your manuscript for publication in the next available issue of EMBO reports. Thank you for your contribution to our journal.

Corresponding Author Name: Thomas Meier
 Journal Submitted to: EMBO Reports
 Manuscript Number: EMBOR-2016-43374V2